# Vine-Shoots as Enological Additives. A Study of Acute Toxicity and Cytotoxicity

**DOI:** 10.3390/foods10061267

**Published:** 2021-06-02

**Authors:** Cristina Cebrián-Tarancón, Francisco Fernández-Roldán, Rosario Sánchez-Gómez, Rosario Salinas, Silvia Llorens

**Affiliations:** 1Cátedra de Química Agrícola, E.T.S.I. Agrónomos y Montes, Departamento de Ciencia y Tecnología Agroforestal y Genética, Universidad de Castilla-La Mancha, Avda. de España s/n, 02071 Albacete, Spain; cristina.ctarancon@uclm.es (C.C.-T.); francisco.fernandez@lajaraba.es (F.F.-R.); rosario.sgomez@uclm.es (R.S.-G.); 2Pago de la Jaraba, Crta, Nacional 310, km 142, 7, 02600 Villarrobledo, Spain; 3Centro Regional de Investigaciones Biomédicas (CRIB), Department of Medical Sciences, Faculty of Medicine of Albacete, University of Castilla-La Mancha, 02008 Albacete, Spain; silvia.llorens@uclm.es

**Keywords:** Microtox^®^ assay, MTT assay, safety, toxicity, vine-shoots

## Abstract

Toasted vine-shoots have been recently proposed as enological additives that can be used to improve the sensorial profile of wines. However, the possible toxicity of this new winery practice has not been studied so far. The aim of this study was to evaluate the toxicity of Tempranillo, Cencibel, and Cabernet Sauvignon toasted vine-shoots when used in winemaking. First, vine-shoots were characterized in terms of minerals and phenolic and furan compounds, and then their acute toxicity and cytotoxicity were studied using Microtox^®^ and the metabolic reduction of 3-(4,5-dimethylthiazol-2-yl)-2,5-diphenyltetrazolium bromide (MTT) assays. High EC_50_ values were obtained when the Microtox^®^ assay was applied to vine-shoot aqueous extracts, similar to the case of herbal infusions. When the MTT assay was used, a cell viability above 70% was observed in all the wines made with those vine-shoots, and an even greater viability was observed in the case of Cabernet Sauvignon. Therefore, it was concluded that those vine-shoots have no cytotoxic potential.

## 1. Introduction

Vine-shoots are the most abundant residue of vineyards, with around 2 tons/hectare/year produced worldwide during vine pruning [1]. In 2019, the area of vineyards worldwide amounted to nearly 7.5 million hectares, highlighting Spain as the most important vine country [2], with the Castilla-La Mancha region cultivating around 50% of such surface. Therefore, a huge amount of this residue is generated, which should be utilized in alternative ways since it is usually left behind or burned on vineyards, causing significant environmental problems.

Vine-shoots are mainly composed of 94% lignocellulosic polymeric material, 55% cellulose and hemicellulose, and 39% lignin [3]. They also contain a small fraction of minerals and phenolic and volatile compounds, with minerals being the most abundant compounds, among which K and Ca have the highest content (5 g/kg each), depending on the characteristics of the vineyard’s soil [4]. Most studies on vine-shoots have focused on their phenolic composition, which usually reaches an average of 3 g/kg, depending on variety. Volatile compounds represent the smallest proportion of constituents in vine-shoots, which normally do not exceed 0.2 g/kg [5,6,7].

Lignocellulosic composition is a main factor in vine-shoots and is the focus of several exploitation studies. However, its use based on the minority compound fraction is highly limited, although some lignocellulosic materials have been recently used as a potential source of bioactive molecules [1]. In fact, among phenolic compounds, it has been shown that (+)-catechin, (−)-epicatechin, ellagic acid, and *trans*-resveratrol are present in high concentrations, with all of them having antioxidant properties [8] and, in the case of the last one, even a chemo-preventive activity [9].

One of the innovative uses for vine-shoots is in the wine industry, similar to how alternative oak products (chips, cubes, etc.) are used. This allows research to focus on a “circular process” since the resources of the vineyard are returned to the wine. Adding toasted oak wood chips to wine throughout the different winemaking stages is a widespread practice, which was authorized in 2005 by the European Union [10]. Similarly, wines resulting from the addition of toasted fragments of vine-shoots after fermentation showed a better quality in terms of aromatic and polyphenolic composition than those of the corresponding control wines (without the addition of vine-shoots) [11].

Throughout the vegetative annual cycle of vines, and mainly during the growing season, a large number of phytosanitary treatments are applied to vines, especially fungicides [12]. Given that vine-shoots have been proposed to be used as enological additives, the presence of fungicide residues should be controlled. Recently, Cebrián-Tarancón [13] demonstrated the dissipation of four of the most used fungicides in Spain into vineyards after the processing of vine-shoots (storage, toasting), which were applied under critical agricultural practices. In addition to the effects of fungicide residues, it is important to focus on toxicity since this aspect has not yet been elucidated, and this type of test is regarded as a crucial step prior to the introduction of any new product to the market.

According to the characterization of vine-shoots, their possible toxicity may be mainly related to their phenolic compounds, mineral composition, or volatile compounds. Generally, phenolic compounds exhibit antioxidant properties [8,9], as indicated above. However, some phenolic compounds have shown some type of toxicity; for example, some flavonoids, such as quercetin, could interfere in essential biochemical pathways [14]. Therefore, it is important to estimate and learn their potential toxicity in vine-shoots [14,15,16]. With regard to mineral composition, specifically heavy metals, their migration in the soil–grape system has been demonstrated [17], and some of them have been found to be present in vine-shoots [4,18]. Finally, in relation to volatile compounds, furans have been proposed to be novel harmful substances in foods that undergo thermal treatment [19]. They originate from cellulose and hemicellulose and increase in vine-shoots during the toasting procedure.

To study the acute toxicity of different types of samples, a modern and cost-effective test, called Microtox^®^, which is based on the inhibition of *Vibrio fischeri* bioluminescence, has proved to be a reliable and sensitive method for the evaluation of the toxicity of herbal infusions [20,21]. As a cytotoxicity assay, 3-(4,5-dimethylthiazol-2-yl)-2,5-diphenyltetrazolium bromide (MTT) is considered to be the most commonly used method for monitoring mitochondrial activity [22], which is an ideal biomarker of chemical-compound-induced cellular damage since the mitochondria play a fundamental role in the initiation and perpetuation of cytotoxicity [23].

As a cellular system, 3T3-L1, a nontumorigenic mouse fibroblast-like cell line was selected because of its wide use [24]. This cell line has been shown to be sensitive to chemical-compound-induced cytotoxicity and does not display an altered cell death potential, avoiding any interference with the testing outcomes [23]. A cytotoxicity study of winery by-products, including vine-shoots, on mitochondrial functions using tumorogenic cells was recently carried out, although the authors indicate that new studies are necessary in this regard, as their study was not conclusive [25].

Since the use of vine-shoots as enological additives is still a very new practice, so far there are no toxicological studies demonstrating its safety. Therefore, to help demonstrate the safety of toasted vine-shoots for use in winemaking, the aim of this study was to evaluate the acute toxicity and cytotoxicity of vine-shoots. For this purpose, extracts and wines prepared with vine-shoots from three of the main red varieties (i.e., Tempranillo, Cencibel, and Cabernet Sauvignon) from the Spanish region of Castilla-La Mancha were evaluated using Microtox^®^ and MTT assays.

## 2. Materials and Methods

### 2.1. Plant Material

Vine-shoots were pruned in February 2020 from three red vinifera cultivars from the Pago de La Jaraba winery (Castilla-La Mancha, Spain): Tempranillo (T; VIVC: 12350), Cencibel (C; a Tempranillo clone adapted to the study area), and Cabernet Sauvignon (CS; VIVC: 1929). The grapevines were planted as a vertical shoot position trellis, pruned to bilateral cordon, and grown in an ecological system under non-irrigation conditions. After pruning, the samples were stored intact in the dark at room temperature (18 ± 3 °C) for 6 months and then ground using a hammer miller (Skid Sinte 1000; LARUS Impianti, Zamora, Spain) to a particle size ranging from 2 mm to 2 cm. Then, they were subjected to a toasting process in an oven with air circulation (Heraeus T6; Heraeus, Hanau, Germany) at 180 °C for 45 min, according to the Cebrián-Tarancón method [6].

### 2.2. Samples: Vine-Shoot Extracts and Wines

#### 2.2.1. Vine-Shoot Extracts

Two different extracts (e) were prepared with the toasted vine-shoots—one with deionized water (W) as control and the other with ethanol/water solution at 12.0% (*v*/*v*; E)—to simulate the contact of vine-shoot pieces in a model wine solution as an enological additive, according to the study by Cebrián-Tarancón [26]. In both cases, vine-shoots with a concentration of 24 g/L were used, which is twice the maximum concentration used in maceration model wines, according to the Cebrián-Tarancón procedure [26], to simulate a more unfavorable situation. For extraction, which was conducted in duplicate for each vine-shoot variety and extractant, maceration was performed in the dark at room temperature, followed by stirring at 800 rpm for 72 h. A total of 12 extracts were obtained, which were filtered using filter paper and then stored at −22 °C until their analysis. The following extracts were obtained: Cencibel aqueous extract (CeW), Cencibel ethanol/water solution extract (CeE), Cabernet Sauvignon aqueous extract (CSeW), Cabernet Sauvignon ethanol/water solution extract (CSeE), Tempranillo aqueous extract (TeW), and Tempranillo ethanol/water solution extract (TeE).

#### 2.2.2. Wines

Wines (w) from Tempranillo, Cencibel, and Cabernet Sauvignon cultivars were made in duplicate according to the classical red winemaking process. After alcoholic fermentation, toasted granulated vine-shoots were added to their corresponding wines, prepared as outlined in Section 2.1. Each wine was placed in contact with its respective vine-shoots, according to the Cebrián-Tarancón method [11]. Therefore, the following wines were obtained: CwC, CSwCS, and TwT. As a control, wines without the addition of vine-shoots were also obtained: Cw, CSw, and Tw. Subsequently, the enological parameters of both the wines that had been in contact with vine-shoots and the control wines were evaluated according to official European methods [27].

### 2.3. Chemical Analysis for Vine-Shoot Characterization

#### 2.3.1. Vine-Shoot Extraction

Before the chemical analysis of the toasted vine-shoots, an extraction step was performed according to Cebrián’s method [5]. Briefly, 20 g of vine-shoots was ground and toasted as outlined in Section 2.1 and then moisturized with 100 g of ethanol/water solution (12.5%, pH 3.62) for 8 h at room temperature. Then, another 100 g of the same solution was added before extraction, which was performed using a microwave NEOS device (Milestone Srl, Sorisole, BG, Italy). Extraction was performed at 75 °C (600 W) for 12 min under reflux to prevent dryness. Then, the extract was centrifuged at 4000 rpm for 10 min, and the supernatant was separated. Subsequently, the solid sample was extracted twice until exhaustion, using the same volume of ethanolic solution (100 mL). The three extracts obtained were mixed and kept at 5–7 °C until their analysis. All extraction procedures were performed in duplicate, hence yielding a total of six extracts.

#### 2.3.2. Phenolic Composition

Individual phenolic compounds were determined according to the Cebrián-Tarancón method [11]. For this purpose, the six extracts obtained according to Section 2.3.1 were injected into an Agilent 1200 HPLC chromatograph (Agilent Technologies, Palo Alto, CA, USA) equipped with a Diode Array Detector (DAD) (Agilent G1315D) coupled to an Agilent ChemStation (version B.03.01) data-processing station. Separation was performed on a reverse-phase ACE Excel 3 C18-PFP (4.6 × 150 mm, 3 μm particle size) and a precolumn ACE Excel UHPLC Pre-Column Filter 1PK (0.5 μm particle size) at 30 °C. The solvents used in the high-performance liquid chromatography (HPLC) were water/formic acid/acetonitrile (97.5:1.5:1, *v*/*v*/*v*; solvent A) and acetonitrile/formic acid/solvent A (78.5:1.5:20, *v*/*v*/*v*; solvent B). The elution gradient was set up for solvent B as follows: 0 min, 5%; 8.40 min, 5%; 12.5 min, 10%; 19 min, 15%; 29 min, 16%; 30 min, 16.5%; 34.8 min, 18%; 37.2 min, 32%; 42 min, 62%; 52 min, 90%; 54 min, 100%; 56 min, 100%; 60 min, 5%; 65 min, 5%. The loop volume was 20 µL. For all the compounds, detection was performed using a DAD by comparison with the corresponding ultraviolet–visible (UV–Vis) spectra and retention time of their pure standards (Sigma-Aldrich, Steinheim, Germany). Compounds were then quantified and identified at different wavelengths: (+)-catechin, (-)-epicatechin, gallic acid, and protocatechuic acid at 280 nm; ellagic acid at 256 nm; *trans*-caftaric acid, piceatannol, and *trans*-ε-viniferin at 324 nm; *trans*-*p*-coumaric acid, *trans-p*-coutaric acid, piceid-*trans*-resveratrol (*trans*-resveratrol-3-glucoside), and *trans*-resveratrol at 308 nm; and quercetin at 365 nm. Quantification was based on calibration curves of the respective standards at five different concentrations achieved by a UV–Vis signal (0.40–260 mg/L, R^2^ = 0.96–1.00). All analyses were performed in duplicate.

#### 2.3.3. Furan Composition

The six extracts obtained according to Section 2.3.1 were extracted using headspace sorptive and stir bar extraction (HS-SBSE) and analyzed by gas chromatography (GC) according to the method of Sánchez-Gómez [18]. The extraction was carried out with 22 mLc of extract stirred at 500 rpm during 60 min. An automated thermal desorption unit (TDU; Gerstel, Mülheim an der Ruhr, Germany) mounted on an Agilent 7890A gas chromatograph coupled to an Agilent 5975C quadrupole electron ionization mass spectrometric detector (Agilent Technologies) equipped with a fused silica capillary column (BP21 stationary phase, 30 m length, 0.25 mm I.D., 0.25 µm film thickness; SGE, Ringwood, Australia) was used. The carrier gas used was helium, with a constant column pressure of 20.75 psi. Mass spectrometry (MS) data acquisition was performed in positive scan mode. To avoid matrix interference, MS quantification was performed in single-ion monitoring mode using characteristic *m*/*z* values. Identification of the compounds was performed using the NIST library and confirmed by comparison with the mass spectra and retention time of their standards. The standards employed were purchased from Sigma-Aldrich (the numbers in parentheses indicate the *m*/*z* values used for quantification): 2-furanmethanol (*m*/*z* = 98), furfural (*m*/*z* = 96), 5-hydroxymethylfurfural (*m*/*z* = 97), and 5-methylfurfural (*m*/*z* = 110). Here, 3-methyl-1-pentanol was used as the internal standard. Quantification was based on calibration curves of the respective standards at five different concentrations (0.5–200 mg/L; R^2^ = 0.97–0.99). All analyses were performed in triplicate.

#### 2.3.4. Mineral Composition

In this study, the following minerals were analyzed: Al, As, Be, Bi, B, Ca, Cd, Co, Cr, Cu, Fe, K, La, Li, Mg, Mn, Mo, Na, Ni, Pb, Rb, Sb, Se, Si, Sr, Ti, Tl, V, and Zn. These minerals were quantified in toasted vine-shoots by inductively coupled plasma optical emission spectrometry (ICP-OES) using an inductively coupled plasma spectrometer (iCAP 6500 Duo; Thermo Fisher Scientific, Madrid, Spain). For digestion, 6 mL of a freshly prepared mixture of HNO_3_ and H_2_O_2_ (5:1, *v*/*v*) was added to 0.5 g of ground vine-shoots (as detailed in Section 2.1) and diluted up to 25 mL with distilled water.

Quantification was based on calibration curves of the respective standards (Sigma-Aldrich) at five different concentrations (0.01–25.00 mg/L; R^2^ > 0.99). All analyses were performed in triplicate.

### 2.4. Toxicity Tests

#### 2.4.1. Microtox^®^ Assay for Vine-Shoot Extracts

Acute toxicity was estimated by determining the bioluminescence inhibition of the marine Gram-negative bacterium *V. fischeri* after 15 min of exposure to the different extracts, according to [28]. The bacteria were reconstituted with a nontoxic 2% NaCl solution and incubated for 20 min at 5.5 ± 1 °C. The light emitted by the bacteria in contact with the samples was analyzed using a Microtox^®^ M500 (AZUR Environmental, Carlsbad, CA, USA), and the results obtained were processed using the MTX-Microtox^®^ program. Toxicity was estimated from the EC_50_ parameter, which expresses the concentration (in mg/mL) of extract that inhibits 50% of bioluminescence. This method is normally used in aqueous samples, and it is not usually used for wines because of the interference that occurs with the red color of wines; therefore, it was used only for extracts [29].

#### 2.4.2. MTT Assay for Vine-Shoot Extracts and Wines

The MTT assay is a colorimetric test that is used to determine the viability of cells via metabolic activity. It is based on the metabolic reduction of 3-(4,5-dimethylthiazol-2-yl)-2,5-diphenyltetrazolium bromide (MTT) to formazan, which is mediated by the mitochondrial enzyme oxidoreductase succinate dehydrogenase, thus reflecting the mitochondrial activity of the cells and, consequently, the number of viable cells present [30]. Yellow water-soluble MTT is metabolically reduced in viable cells to purple-colored formazan crystals.

To perform the MTT assay, embryonic 3T3-L1 fibroblasts (American Type Culture Collection, Manassas, VA, USA) were cultured as per the supplier’s instructions. In all the experiments, cells were used within the fourth passage. The cells were grown on 96-well sterile plates at a cellular density of 2 × 10^3^ cells/well in a cell culture medium (Dulbecco’s modified Eagle’s medium (DMEM), 90%), supplemented with L-glutamine (1%), penicillin/streptomycin (0.5%), and inactivated fetal bovine serum (FBS, 10%), at 37 °C in a 5% CO_2_ incubator. When the cells reached optimum confluence (70–80%), the study samples replaced the culture medium.

The MTT viability assay was performed as previously described in the study by Escobar [30], with a slight modification. An MTT stock solution in sterile phosphate-buffered saline (PBS, 5 mg/mL) was freshly prepared and filtered using syringe filters (<0.2 μm pore size; Nalgene™ Syringe Filters; Thermo Fisher Scientific, Waltham, MA USA). Working MTT solution was diluted 1:10 in DMEM without phenol red (without PBS). The medium was then removed from the culture plates, and the cells were washed with DMEM without phenol red. Then, the working MTT solution (100 μL) was deposited in each well, and the cell culture plates were incubated for 3 h under standard culture conditions. Then, the MTT solution was carefully removed, and the cells were washed with PBS. Finally, to solubilize the formazan crystals that were formed, 100 μL of dimethyl sulfoxide (DMSO) was added to each well, followed by stirring at room temperature for 10–15 min, yielding a purple-colored solution. After solubilization, the solutions were transferred to fresh 96-well plates, and absorbance at an optical density (OD) wavelength of 595 nm was measured using spectrophotometry (Asys UVM 340; Microplate Readers, Cambridge, UK) [31]. In general, the number of viable cells correlates with the color intensity determined. Data obtained from at least three replicates in each experimental condition from two independent experiments were used for analysis. To calculate the cellular viability, the following equation was used:%Viability = 100 × 〖OD〗(595a)/〖OD〗595b(1)
where OD_595a_ is the color intensity of the problem assay, and OD_595b_ is the reference. All values become final ODs after the subtraction of background absorbance (wells without cells but exposed to MTT, washed, and exposed to DMSO).

First, the cytotoxicity of the ethanol solvent at 12% was checked. For this purpose, concentration–response curves were created by dilution in the range of 1:10^3^ to 1:10^6^ to prevent the final ethanol concentration in the wells from exceeding 0.1% (*v*/*v*). The effect of this solvent on the viability of 3T3-L1 was measured by the MTT assay using Equation (1), where OD_595a_ is the average OD of wells corresponding to the ethanol solvent and OD_595b_ is the average OD of all wells of untreated control cells (C-control). The absence of cytotoxicity (100% viability) was attributed to the C-control.

Moreover, the cytotoxicity of vine-shoot ethanol/water extracts and the dilution of vine-shoot ethanol/water (12%) extracts were assessed, for which 50% viability was determined. For this purpose, respective dose–response curves were created. Viability was also determined using Equation (1), where OD_595a_ is the average OD of wells corresponding to vine-shoot extracts (CeE, CSeE, or TeE), and OD_595b_ is the average OD of wells corresponding to ethanol solvent at 12% (1:10^3^). The absence of cytotoxicity (100% viability) was attributed to the control ethanol solvent at 12%.

It should be noted that the dilution of the vine-shoots that produced 50% 3T3-L1 viability was 1:10^4^ for all vine-shoot extracts tested. Therefore, dilution of control wines and wines with vine-shoots was performed in the range of 1:10^3^ to 1:10^6^.

Finally, the cytotoxicity of wines with vine-shoots was assessed. Concentration–response curves of all wines (Tw, Cw, CSw, TwT, CwC, and CSwCS) were created by dilution in the range of 1:10^3^ to 1:10^6^. Different dilutions were prepared in a cell culture medium at the beginning of each experiment, and the cells were further incubated under cell culture conditions for 72 h. Three controls were included on the same plates as follows: C-control, ethanol solvent at 12% (cells treated with 12% ethanol/water solution, simulating the highest alcoholic degree of wines), and vine-shoot extracts (cells macerated with vine-shoots, CeE, CSeE, and TeE, in 12% ethanol/water solution). Viability was determined using the same previously mentioned equation, where OD_595a_ is the average OD of wells corresponding to control wines (Cw, CSw, or Tw) or those with vine-shoots (CwC, CSwCS, or TwT), and OD_595b_ is the average OD of all wells of C-control. The absence of cytotoxicity (100% viability) was attributed to the C-control. The results are presented as a percentage of the following controls: C-control for Tw, Cw, CSw, TwT, CwC, and CSwCS and ethanol solvent at 12% for CeE, CSeE, and TeE.

### 2.5. Statistical Analysis

The descriptive analysis results regarding vine-shoot parameters composition and toxicity results (Microtox^®^ and MTT assays) were analyzed by one-way analysis of variance (ANOVA) at 95% probability level according to Fisher’s least significant difference (LSD) to determine the differences among the samples. Additionally, a multivariate analysis of variance (MANOVA) was performed with the purpose of having an overall view of the influence that variety and extractant factors have in the Microtox^®^ assays and that wine type and the addition of vine-shoots factors have in the MTT assay. Principal component analysis (PCA) was performed to summarize the results for the vine-shoots composition and Pearson’s correlation analysis was summarized to investigate the relationship between toxicity results (Microtox^®^ and MTT assays) and analyzed compounds. All statistical analyses were conducted using the Statgraphics Centurion statistical program (version 18.1.12; StatPoint, Inc., The Plains, VA, USA).

## 3. Results and Discussion

### 3.1. Vine-Shoot Composition

The chemical compounds capable of being transferred from vine-shoots to aqueous and ethanolic solutions are, according to research by Cebrián-Tarancón [11,26], some phenolics, furans, and minerals, which are shown in Table 1.

In that study, the authors also concluded that this transfer varied depending on the type of compound and matrix and that the transferred proportion was very small but sufficient to influence the final quality of the wines.

It can be seen that the most abundant compounds were minerals (around 75%), followed by phenolic (around 20%) and furan (around 1%) compounds. As is already known, flavanols were found to be highly abundant in the group of phenolic compounds, constituting around 65% of the phenolic fraction in the samples studied, with the amount of (−)-epicatechin being twice that of (+)-catechin. In addition, ellagic acid and *trans*-resveratrol were found to constitute approximately 19% and 5%, respectively, of the total phenolic compounds. It can be observed that the Cencibel vine-shoots had a higher total phenolic content than that of Cabernet Sauvignon and Tempranillo. Cencibel was found to exhibit the highest content of flavanols and phenolic acids, and Cabernet Sauvignon was found to exhibit the highest content of total stilbenes, although ε-viniferine was found to have a higher concentration in Cencibel. It should be noted that no quercetin, a flavonol, was found in the Tempranillo vine-shoots. Furan compounds are originally present in vine-shoots and increase in concentration with toasting. They were found to be more abundant in Cabernet Sauvignon, with the lowest content found in Tempranillo, 2-furanmethanol being the only one found in the three varieties in similar concentrations. Other furans were found to have significantly lower concentrations in Tempranillo vine-shoots, although 5-hydroxymethylfurfural had a similar concentration in Cencibel and an almost doubled concentration in Cabernet Sauvignon. Finally, the total mineral content observed was mainly due to the high values of K, Ca, Mg, and Na, which are the main macronutrients of vines and are found in significantly high concentrations in Cencibel and low concentrations in Cabernet Sauvignon vine-shoots. Minority minerals were also found to constitute the highest total content in Cencibel, similar to Tempranillo and Cabernet Sauvignon vine-shoots. However, some metals, such as Cr, are found only in Cencibel, whereas Bi and Pb are not found in Cabernet Sauvignon, while Mn and Co are found in significantly high concentrations in Cabernet Sauvignon vine-shoots. All of this suggests that the soil in which the three vine varieties are grown provides a different mineral profile, given that they were all cultivated in the same way.

The results outlined in Table 1, grouped into organic compounds (phenolics, furans) and mineral compounds, were subjected to separate principal component analysis (PCA), as shown in Figure 1 and Figure 2, respectively. In both cases, it can be observed that there was favorable separation of the three varieties of vine-shoots using two statistical functions. In the case of organic compounds, the first component explained 41.67% of the variance and the second 30.94%, with ε-viniferine, ellagic acid, and gallic acid being the variables with the greatest weight in component 1 and piceatannol, 5-hydroxymethylfurfural, and 2-furancarboxaldehyde being those in component 2 (Figure 1). In the case of minerals, the separation was even better, with component 1 explaining 60.76% of the variance and component 2 explaining 26.18% of it (with Fe, Na, and K being the variables with the greatest discriminant capacity in component 1 and Mn, Tl, and Sb being the ones with the greatest weight in component 2; Figure 2). Therefore, it can be concluded that the three varieties of vine-shoots are different in terms of their chemical composition and may exhibit different behaviors in toxicity tests.

### 3.2. Toxicity Results

#### 3.2.1. Vine−Shoot Acute Toxicity Evaluated toward *V.*
*fischeri*

Table 2 shows the results of the Microtox^®^ assay in terms of EC_50_. Generally, EC_50_ represents the effective lethal concentration (in mg/mL) that corresponds to the proportion of extract that causes mortality or inhibition of 50% of the exposed bacteria (*V.*
*fischeri*). When all the extracts were compared, significant differences were observed among them. It is worth mentioning that the extracts obtained from the Tempranillo variety were those that showed the highest (TeE) and lowest (TeW) values of EC_50_, whereas the rest of the extracts showed mean values within this interval. With regard to each variety, it can be observed that, for the Tempranillo and Cabernet Sauvignon vine−shoots, the ethanol/water solution extracts had high EC_50_ values, although such differences were significant only for Tempranillo. This indicates that a larger amount of ethanol/water extracts was necessary to reduce the bioluminescence of the bacteria by half since the higher the EC_50_ value, the lower the toxicity. Therefore, as previously mentioned, the lowest values were found for the aqueous extract of Tempranillo, hence making this extract the most effective one of all extracts against *V. fischeri.*

In view of these results, and with the aim of highlighting the influence of each factor studied (vine−shoot variety and extractant) on EC_50_, a multivariate analysis of variance (MANOVA) was performed. The results indicated that the extractant (water or 12% ethanol/water solution) selected was the most influential factor on acute toxicity, with a *p*-value of 0.0052, whereas for the variety factor, the significance value was only *p* < 0.1. However, the results of the interaction between both factors were significant (*p* = 0.0122), which is explained by the different behaviors of the three varieties (Tempranillo, Cencibel, and Cabernet Sauvignon) in relation to the extractant used.

In addition to using water as an extractant, a 12% ethanol/water solution was selected for extraction to simulate a model wine solution, which is a way to determine what might happen in actual wines since this assay cannot be performed because of the samples’ red color [29]. Therefore, under this premise, and in an attempt to extrapolate these results to real wines, using vine−shoots as enological additives does not present an acute toxicity problem. However, in the trial with MTT that followed, an in−depth toxicological study on vine−shoots was performed.

To our knowledge, no studies in the literature have focused on performing comparisons on the acute toxicity of vine−shoots. Moreover, in the case of oak wood, the material that is most used for aging wines, studies related to toxicity have not focused on this type of assay [32]. However, since one of the extractants considered was water, some comparisons have been made with the results obtained for aqueous infusions of aromatic plants. Skotti’s research [20] evaluated the toxicity of herbal infusions at *V. fischeri* using bioluminescence inhibition (Microtox^®^) of several aqueous extracts from different Greek medicinal and aromatic plants. The EC_50_ results were found to be notably higher in oregano (*Origanum vulgare* L.) infusions in boiled water and in dittany (*Origanum dictamnus* L.) infusions at room temperature than those in vine−shoots, suggesting that the extracts from vine−shoots (aqueous or aqueous/ethanolic) had less acute toxicity. Furthermore, in the present study, the proportion of vine−shoots was 24 g/L, which is 2.4 times higher than the concentration used in the aforementioned study, in which a concentration of 2 g/200 mL of plants was used. Since the extracts were tested without removing any part of the vine−shoots, the reduction of the EC_50_ values, which as indicated above does not imply toxicity, should be related to some of the extracted compounds. In this case, and given the composition importance, it was focused on phenolic, volatile, and mineral compounds. Although phenolic compounds are mainly associated with health benefits, some reports highlight side effects and toxicity associated with phenolics [14,15]. Therefore, different correlations were performed in an attempt to establish a relationship between EC_50_ values and the phenolic compounds analyzed in vine−shoots; however, no correlation was established (the R values were less than 0.05; data not shown). Similar results were found for herbal infusions when no correlation between bioluminescence inhibition and total phenolic content was found [20,21].

In terms of the volatile composition of vine−shoots, only furan compounds were included in the present study because of their possible implications in toxicity since the contents of the rest of the volatile compounds quantified in vine−shoots were found to be significantly lower than furan [6]. This group of compounds should be taken into account when considering the content of cellulose and hemicellulose in vine−shoots since temperature enhances the degradation of sugars released from vine−shoots wood and promotes the formation of furans. Although the contribution of furan compounds, such as furfural and 5−hydroxymethylfurfural (HMF), to the aroma and flavor of food is well known, some research facilities and international organizations worldwide (e.g., U.S. Food and Drug Administration) have regarded furans as novel harmful substances in foods that undergo thermal treatment. Thus, from a safety perspective and for food quality assurance, EC Regulation [33] sets up a maximum limit for HMF of 25 mg/kg in concentrated rectified grape must. On this premise, it is important to control the content of furans in vine−shoots to properly exploit them as enological additives. With regard to the concentrations of vine−shoot extracts and the results related to the Microtox^®^ assay, the highest EC_50_ values did not correspond to the highest content of furan compounds, given that the toxicity value was dependent on the extractant used. In fact, no correlations were found among furan compounds and EC_50_ values.

#### 3.2.2. Cytotoxicity of Vine−Shoots and Wines Macerated with Them, Evaluated through 3T3−L1 Cell Viability

As indicated in Section 2, for the MTT assay related to cytotoxicity, some tests were performed. Before assessing the in vitro cytotoxicity, it was necessary to establish the cellular system, cytotoxicity assay, and exposure conditions. In all tests, according to ISO 10993-5 [34], it was considered that a tested product has a cytotoxic potential when the cell culture viability decreased to <70% in comparison to Group b (untreated control cells and ethanol solvent at 12%, 1:10^3^) (reference assay; OD_595b_; see Equation (1)), which was set at 100% viability. It was also decided that a 72 h exposure period is preferable to exposure for 24 or 48 h since this would allow more time for vine−shoots to exert their potential toxic effects [35].

The ethanol control solvent diluted 1:103 produced a slight decrease in viability (97.3% ± 6.4%), although this decrease was not significantly different with respect to C−control. Regarding the vine−shoot ethanol/water (12%) extracts (CeE, CSeE, and TeE), it was observed that the dilution of these extracts, which produced 50% viability, was as follows: TeE, 4.74 × 10^−4^ and R^2^ = 0.9765; CeE, 8.24 × 10^−4^ and R^2^ = 0.9305; CSeE, 3.24 × 10^−4^ and R^2^ = 0.9704. Moreover, the percentages of viability that produced the highest concentration tested (1:10^3^) were 92.4% ± 15.3%, 99.7% ± 10.1%, and 87.3% ± 8.1% for TeE, CeE, and CSeE, respectively, which did not affect the cellular viability compared with the 12% ethanol solvent control at the same dilution or among the different extracts. The viability produced by each of them showed slight differences, in the order CSeE > TeE > CeE. This variation may be caused by the grape variety; indeed, as some studies have reported, vine−shoot extracts from other varieties such as Riesling showed cytotoxic effects [25]. For wines, the results obtained by evaluating the effects of vine−shoot maceration on 3T3−L1 viability are shown in Figure 3: Tempranillo (a), Cencibel (b), and Cabernet Sauvignon (c). The figure also shows the maximum viability detected for C−control (dashed line, set to 100%) and percentage of viability for TeE, CeE, and CSeE. Moreover, for each wine type (Tempranillo, Cencibel, and Cabernet Sauvignon) and each dilution studied, the cells’ relative viabilities when exposed to wines (control wines and wines elaborated with vine−shoots) are represented. In the case of the highest dilution, the results of vine−shoot ethanol/water (12%) extracts were also included to facilitate visual comparison.

Albeit with slight differences, the profiles shown by the three types of wines were similar. In general, the percentages of viability for all wine dilutions were greater than 85%, a value that is greater than 70%, which indicates non−cytotoxicity. It should be noted that the viability of 3T3−L1 in the presence of wines macerated with vine−shoots did not exhibit statistically significant differences compared with their corresponding control wines. Moreover, no statistically significant differences were observed when the percentage of viability of wines (control wines and wines with vine−shoots) was compared to that of the 12% ethanol solvent or its corresponding vine−shoot extracts at the highest concentration, corresponding to 1:10^3^ dilution. In comparison to C−control (set at 100% viability), wines from Tempranillo (Tw, TwT) seemed to slightly reduce the viability since at the highest concentration these values were 11.5% and 11.0%, respectively. For Cencibel wines (Cw, CwC), it was observed that this highest concentration tested induced only 10.3% and 9.7% mortality, respectively. Relative to the Cabernet Sauvignon wines (CSw, CSwCS), these percentages were reduced by 11.5% and 10.7%, respectively. However, in all cases, the viability exceeded 70% in comparison with C−control, as mentioned above. This value is considered the cutoff for designating wines as nontoxic, according to ISO norms [34]. Although the results were similar for the three wines studied, it can be observed that there is a trend toward a lower reduction in viability associated with Cabernet Sauvignon wines with their own vine−shoots (CSwCS), followed by Cencibel (CwC) and Tempranillo (TwT) wines.

Investigating further the two factors considered in this assay (i.e., wine type and addition of vine−shoots), MANOVA showed that neither of these two factors exhibited significant differences (*p* > 0.05; data not shown). Zhand [36] evaluated the effect of *trans*−resveratrol−spiked grape skin extracts at different concentrations on 3T3−L1 cell viability and found no toxicity. In addition, no dose−dependent effect of trans−resveratrol was observed at extract concentrations above 500 μg/mL. It should be noted that the wines investigated in the present study showed a mean concentration of such a compound of 2 mg/L, and the results obtained with the MTT assay are in agreement with previous studies. Recently, in the studies of Medrano−Pidal [16,37] related to the cytotoxicity of some stilbenes and a stilbene extract enriched in *trans*−resveratrol and *trans*−ε−viniferin, a significant decrease in the viability of both human intestinal Caco−2 and liver Hep−G2 cell lines was found after exposure with *trans*−resveratrol; however, the stilbenes and enriched stilbene extract presented a lower effect.

## 4. Conclusions

This study, which assessed the potential toxicity of vine−shoots used as enological additives, shows that no acute toxicity was observed when a Microtox^®^ assay was performed on the extracts obtained from the three varieties studied (Tempranillo, Cencibel, and Cabernet Sauvignon). In relation to wines obtained with the addition of their own vine−shoots, no cytotoxic effect was observed in 3T3−L1 fibroblast cells exposed for 72 h to wines. However, the viability produced by exposure with vine−shoot extracts was lower but did not exhibit a cytotoxic potential either. Therefore, all of these results suggest that vine−shoots can be used as enological additives and that wines with their own vine−shoots added are probably safe for consumption.

## Figures and Tables

**Figure 1 foods-10-01267-f001:**
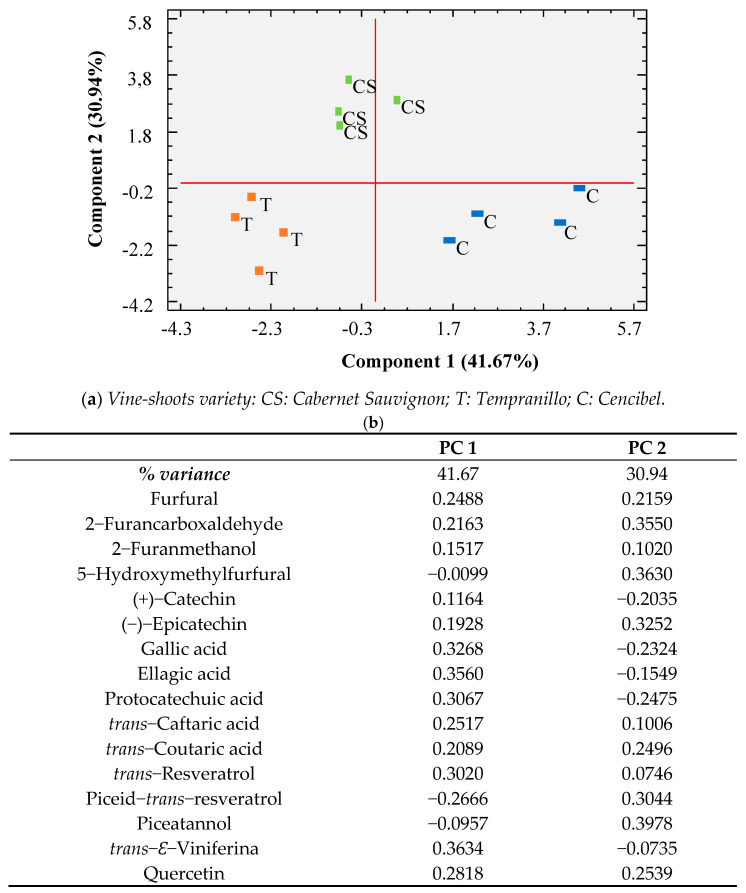
Principal component analysis (PCA) results carried out with phenolic and furans composition. (**a**) Projection of wine samples in the plane formed by the two main components; (**b**) weights of the variables in the first two principal components.

**Figure 2 foods-10-01267-f002:**
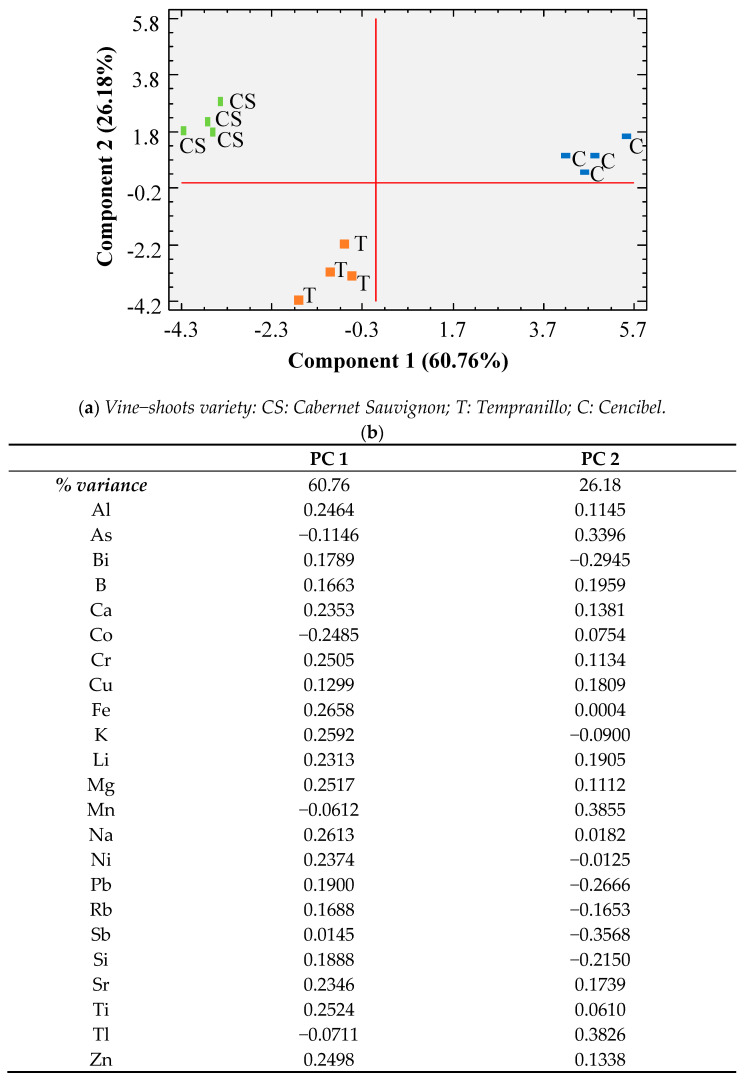
Principal component analysis (PCA) results carried out with mineral composition. (**a**) Projection of wine samples in the plane formed by the two main components; (**b**) weights of the variables in the first two principal components.

**Figure 3 foods-10-01267-f003:**
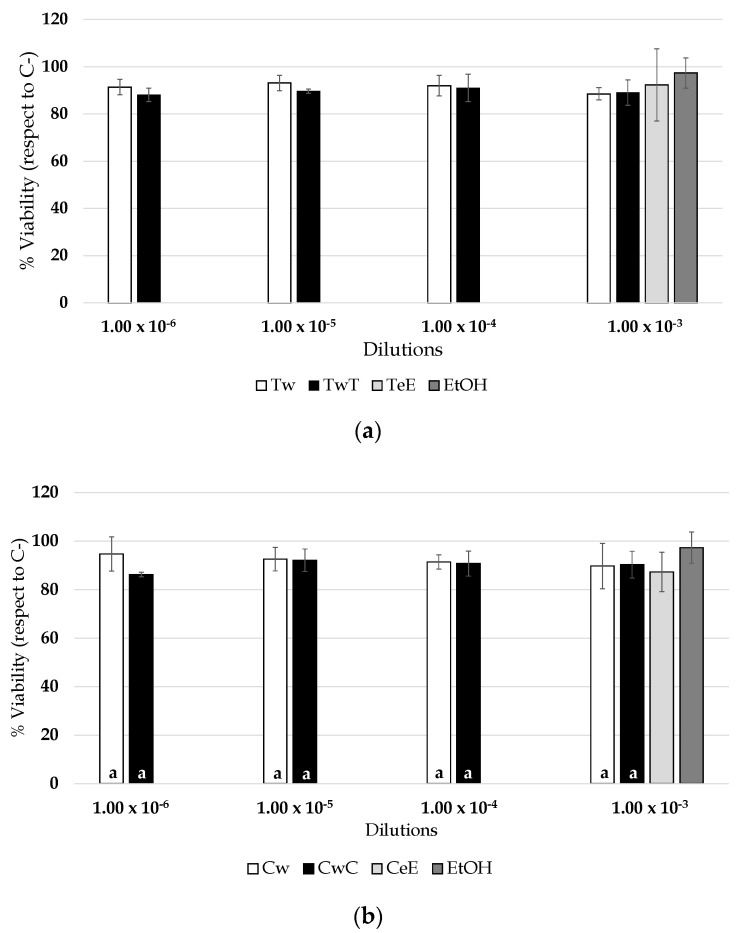
Effect on cellular viability of: (**a**) Tempranillo wines, with or without addition of their own vine−shoots; (**b**) Cencibel wines, with or without addition of their own vine−shoots; (**c**) Cabernet Sauvignon wines, with or without addition of their own vine−shoots. Different letters indicate significant differences among wines of the same variety according to the Fisher’s LSD test (α < 0.05).

**Table 1 foods-10-01267-t001:** Vine-shoots composition.

	Tempranillo	Cencibel	Cabernet Sauvignon	F ^1^
*Phenolic compounds (mg*/*kg)*				
*Flavanols*				
(+)-Catechin	964.68 ± 177.92 a	1166.22 ± 261.08 a	932.40 ± 38.58 a	1.90
(−)-Epicatechin	1595.66 ± 260.68 a	1950.65 ± 191.75 b	2107.58 ± 66.98 b	7.56 *
*Acids*				
Gallic acid	17.62 ± 3.19 a	30.63 ± 3.63 b	15.13 ± 0.30 a	35.57 ***
Ellagic acid	790.90 ± 34.84 a	937.03 ± 51.71 b	795.32 ± 11.06 a	20.67 ***
Protocatechuic acid	11.03 ± 2.32 a	19.18 ± 3.84 b	8.19 ± 1.15 a	18.14 ***
*trans*-Caftaric acid	52.21 ± 14.51 a	74.37 ± 26.26 a	64.28 ± 1.63 a	1.64
*trans*-Coutaric acid	10.13 ± 1.59 a	14.29 ± 2.60 b	14.77 ± 3.20 b	4.00 *
*Stilbens*				
*trans*-Resveratrol	200.43 ± 13.84 a	248.33 ± 6.99 b	235.49 ± 28.38 b	7.05 *
Piceid-*trans*-resveratrol	22.79 ± 0.04 b	15.72 ± 0.13 a	26.87 ± 0.78 c	608.21 ***
Piceatannol	82.67 ± 8.64 a	80.07 ± 10.91 a	126.46 ± 2.20 b	41.04 ***
trans-ε-viniferin	64.31 ± 0.21 a	187.26 ± 0.66 c	96.86 ± 0.46 b	70,261 ***
*Flavonols*				
Quercetin	n.d.	14.53 ± 0.12 a	15.14 ± 0.28 b	16.23 **
Total	3812.44 ± 6.76 a	4738.28 ± 15.09 c	4438.49 ± 10.64 b	692.36 ***
*Furan compounds (mg*/*kg)*				
Furfural	12.37 ± 3.72 a	18.83 ± 3.35 b	18.87 ± 4.54 b	3.67 *
2-Furancarboxaldehyde	0.57 ± 0.22 a	1.49 ± 0.33 b	1.90 ± 0.44 b	15.45 **
2-Furanmethanol	146.44 ± 25.26 a	165.03 ± 28.85 a	164.71 ± 30.86 a	0.56
5-Hydroxymethylfurfural	16.94 ± 7.17 a	17.16 ± 5.68 a	30.46 ± 9.45 b	4.16 *
Total	176.32 ± 29.61 a	202.51 ± 37.70 a	215.94 ± 19.90 a	1.81
*Minerals (mg*/*kg)*				
Al	5.12 ± 1.88 a	25.03 ± 3.74 b	5.16 ± 0.77 a	65.63 ***
As	1.20 ± 0.24 a	1.59 ± 0.35 a	2.31 ± 0.28 b	11.07 **
Bi	4.76 ± 0.23 a	4.33 ± 0.70 a	n.d.	0.99
B	7.85 ± 1.74 a	10.35 ± 1.22 a	8.44 ± 2.06 a	1.75
Ca (g/kg)	2.71 ± 0.06 a	4.89 ± 0.99 b	2.81 ± 0.23 a	13.34 **
Co	0.08 ± 0.02 b	0.04 ± 0.02 a	0.12 ± 0.02 c	16.00 **
Cr	n.d.	0.16 ± 0.02	n.d.	-
Cu	3.33 ± 0.60 a	4.09 ± 0.08 a	3.62 ± 0.77 a	1.36
Fe	12.15 ± 1.58 b	20.16 ± 1.88 c	8.39 ± 0.81 a	48.89 ***
K (g/kg)	5.96 ± 0.06 b	7.71 ± 0.04 c	4.10 ± 0.37 a	207.72 ***
Li	0.26 ± 0.00 a	0.57 ± 0.05 b	0.32 ± 0.05 a	49.19 ***
Mg (g/kg)	0.56 ± 0.13 a	1.03 ± 0.03 b	0.53 ± 0.03 a	36.92 ***
Mn	13.28 ± 1.63 a	19.73 ± 0.79 b	24.41 ± 1.01 c	65.32 ***
Na	0.03 ± 0.01 b	0.08 ± 0.01 c	0.01 ± 0.01 a	41.55 ***
Ni	0.24 ± 0.05 ab	0.41 ± 0.16 b	0.15 ± 0.03 a	5.55 *
Pb	0.26 ± 0.05 a	0.26 ± 0.06 a	n.d.	0.01
Rb	0.78 ± 0.20 a	0.82 ± 0.06 a	0.60 ± 0.07 a	2.41
Sb	0.35 ± 0.07 b	0.26 ± 0.07 ab	0.21 ± 0.04 a	5.16 *
Si	25.92 ± 5.36 b	26.89 ± 4.70 b	12.13 ± 0.24 a	12.08 **
Sr	69.11 ± 9.85 a	144.91 ± 3.46 b	81.17 ± 11.49 a	61.95 ***
Ti	0.81 ± 0.14 a	1.48 ± 0.01 b	0.67 ± 0.23 a	24.16 **
Tl	0.65 ± 0.15 a	1.20 ± 0.20 b	1.71 ± 0.30 c	16.44 **
Zn	5.77 ± 1.63 a	24.13 ± 0.14 b	6.56 ± 0.52 a	330.92 ***
Total (g/kg)	9.42 ± 0.32 b	14.02 ± 1.24 c	7.60 ± 0.75 a	59.38 ***

For each parameter, different letters indicate significant differences among different vine-shoots varieties according to the Fisher’s LSD test (α < 0.05). Concentrations of Be, Cd, La, Mo, Se, and V were zero. n.d.: no data. ^1^ Significant values are typed in bold according to: * *p* value < 0.05; ** *p* value < 0.01; *** *p* value < 0.001.

**Table 2 foods-10-01267-t002:** Toxicity results from Microtox^®^ assay of vine−shoot extracts as EC_50_ (mg/mL) after 15 min incubation time.

	EC_50_ (mg/mL)	F ^1^	
**TeW**	8.70 ± 0.54 a, A	9.15 **	31.32 *
**TeE**	17.37 ± 2.12 d, B		
**CeW**	11.96 ± 1.70 ab, A		0.04
**CeE**	11.71 ± 0.49 ab, A		
**CSeW**	13.42 ± 2.14 bc, A		2.36
**CseE**	15.75 ± 0.17 cd, A		

TeW: Tempranillo aqueous extract; TeE: Tempranillo ethanol/water solution extract; CeW: Cencibel aqueous extract; CeE: Cencibel ethanol/water solution extract; CSeW: Cabernet Sauvignon aqueous extract; CSeE: Cabernet Sauvignon ethanol/water solution extract. Small letters indicate significant differences among all different extracts from the three varieties considered and capital letters indicate significant differences among the different extracts of each variety according to the Fisher’s LSD test (α < 0.05). ^1^ Significant values are typed in bold according to: * *p* value < 0.05; ** *p* value < 0.01.

## Data Availability

The data presented in this study are available on request from the corresponding author.

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
