# Peer review of "Vine-Shoots as Enological Additives. A Study of Acute Toxicity and Cytotoxicity"

_foods, 2021, doi:10.3390/foods10061267_

Round 1

Reviewer 1 Report

Dear Authors,

my congratulations on excellent work and presentations of your results. There is just one little thing:

On the page 7 of 17 you state that all varetis have been cultivated in the same way and that the soil provide a different mineral profile. It would be useful information if you could provide (in Material and methods chapter) more information about the soils and growing conditions, as well as about the way of growing in briefly.

Author Response

Manuscript ID: FOODS-1226045

RESPONSE TO REVIEWER 1 AND CHANGES WE HAVE MADE TO THE MANUSCRIPT

The paper under revision has its research focus on the toxicity and cytotoxicity of the toasted vine-shoots used as an alternative enological additive. The topic presented fulfils the requirement. Results produced in this paper are of great interest as they enlarge our knowledge on vine-shoots and their application. However, some experimental information needs to be clarified.

Authors would like to thank the reviewer for his/her revision and appropriate suggestions to the paper, enhancing the quality of this research. We hope that the answers outlined below will be satisfactory. All changes in the manuscript are indicated with Microsoft Word's track changes.

My comments and editing follow:

It was difficult to state my thoughts since no line numbering existed, however, I list my comments referring either to page numbers and paragraphs or sections.

Authors apologize for this reason.

Introduction

Page 2, paragraph 3.

“However, some phenolic compounds have shown some type of toxicity.” Please provide more details and references.

This sentence has been rewritten for a better understanding (lines 74-75, page 2).

Materials and Methods

Page 3, section 2.3. Chemical analysis for vine-shoot characterization

The whole section needs adequate clarification.

This section has been revised and changed accordantly.

Why authors proceed with vine-shoot extraction, using three different ways of extraction and then the extracts were mixed? Please clarify.

For vine-shoots characterization, extracts were obtained after an exhaustion of the material, for which it was established to carry out 3 identical extractions and then mix them (section 2.3.1.). Foy toxicity studies, other two extracts (section 2.2.1.) were obtained: a) Water extracts: to be used as a control sample since Microtox® assay is normally performed in aqueous solutions but not in ethanolic solutions; b) ethanol/water solution, at 12.0% (v/v): to simulate the contact of vine-shoot pieces in a model wine solution as an enological additive (Cebrián-Tarancón et al. 2019), in order to determine which could be the behavior in a similar condition than in real wines, since for Microtox® assay cannot be performed in wines because of the samples’ red color.

I think it would be most realistic to analyse the water and ethanol/water extracts of vine-shoots that are prepared above (section 2.1), rather than perform a super exhausting extraction.

Reviewer is right, but the aim was to establish comparisons with the most unfavorable situation (higher compounds content).

2.3.1. Vine-shoot extraction

I assume authors refer to 100 mL not 100 g of ethanol/water solution.

In this case 100 g is correct since the proportion was stablish in terms of weight.

2.3.2. Phenolic composition

Clarify what samples were analysed.

More information related to the number and type of samples has been added for a better understanding (lines 159-160, page 5).

Be careful with superscript (R2).

It has been changed and checked throughout the manuscript.

2.3.3. Furan composition

Clarify what samples were analysed. Provide details on the quantity used.

More information related to the samples analyzed has been added (line 183, page 4). Also, some details about extract method have been included for a better understanding (line 186, page 4).

Be careful with superscript (R2).

It has been changed and checked throughout the manuscript.

2.3.4. Mineral composition

Mineral composition is studied on ground vine-shoots.

The sentence has been rewritten (line 211, page 6).

Be careful with subscript (HNO3 and H2O2), superscript (R2).

It has been changed and checked throughout the manuscript.

2.4.1. Microtox® assay for vine-shoot extracts

  1. fischeri in italics.

It has been changed and checked throughout the manuscript.

Correct EC50 to EC50.

It has been changed and checked throughout the manuscript.

2.4.2. MTT assay for vine-shoot extracts and wines

There is a mesh with superscripts and at first it was difficult to follow the meaning.

Please correct.

It has been changed and checked throughout the manuscript.

Correct CO2.

It has been changed and checked throughout the manuscript.

Delete n from An MTT solution.

It has been corrected (line 245, page 6).

Results and discussion

Pages 6-7

“All of them constituted a soluble fraction… by Cebrián-Tarancón works”. Please clarify and rephrase.

The sentence has been rewritten (lines 323-325, page 7).

Page 12.

Authors state that they tried to establish a relationship between EC50 values and the phenolic compounds analyzed in vine-shoots as reported by Skotti et al., (2011) for herbal infusions. Skotti et al., analysed the phenolic compounds from the same extracts used for Microtox (2g/200mL). Authors provide results from vine-shoots extracts after different extraction procedure and not from the extracts tested in Microtox.

The reviewer is right and, as indicated above, the idea was to study that in the most unfavorable conditions, there was no correlation with this type of compounds.

Reviewer 2 Report

The paper under revision has its research focus on the toxicity and cytotoxicity of the toasted vine-shoots used as an alternative enological additive. The topic presented fulfils the requirement. Results produced in this paper are of great interest as they enlarge our knowledge on vine-shoots and their application. However, some experimental information needs to be clarified.

My comments and editing follow:

It was difficult to state my thoughts since no line numbering existed, however, I list my comments referring either to page numbers and paragraphs or sections.

Introduction

Page 2, paragraph 3.

“However, some phenolic compounds have shown some type of toxicity.” Please provide more details and references.

Materials and Methods

Page 3, section 2.3. Chemical analysis for vine-shoot characterization

The whole section needs adequate clarification.

Why authors proceed with vine-shoot extraction, using three different ways of extraction and then the extracts were mixed? Please clarify.

 I think it would be most realistic to analyse the water and ethanol/water extracts of vine-shoots that are prepared above (section 2.1), rather than perform a super exhausting extraction.

2.3.1. Vine-shoot extraction

I assume authors refer to 100 mL not 100 g of ethanol/water solution.

2.3.2. Phenolic composition

Clarify what samples were analysed.

Be careful with superscript (R2).

2.3.3. Furan composition

Clarify what samples were analysed. Provide details on the quantity used.

Be careful with superscript (R2).

2.3.4. Mineral composition

Mineral composition is studied on ground vine-shoots.

Be careful with subscript (HNO3 and H2O2), superscript (R2).

2.4.1. Microtox® assay for vine-shoot extracts

  1. fischeri in italics.

Correct EC50 to EC50.

2.4.2. MTT assay for vine-shoot extracts and wines

There is a mesh with superscripts and at first it was difficult to follow the meaning. Please correct.

Correct CO2.

Delete n from An MTT solution.

Results and discussion

Pages 6-7

“All of them constituted a soluble fraction… by Cebrián-Tarancón works”. Please clarify and rephrase.

Page 12.

Authors state that they tried to establish a relationship between EC50 values and the phenolic compounds analyzed in vine-shoots as reported by Skotti et al., (2011) for herbal infusions. Skotti et al., analysed the phenolic compounds from the same extracts used for Microtox (2g/200mL). Authors provide results from vine-shoots extracts after different extraction procedure and not from the extracts tested in Microtox.

Author Response

Manuscript ID: FOODS-1226045

RESPONSE TO REVIEWER 2 AND CHANGES WE HAVE MADE TO THE MANUSCRIPT

This is a very interesting research topic. My comments are as follows.

Firstly, please accept our thanks for the comments you have made on our paper, that increase their quality. We hope that the answers outlined below will be satisfactory. All changes in the manuscript are indicated with Microsoft Word's track changes.

  1. A minor suggestion: Perhaps authors can use a colon (:) to connect two sentences.

No changes have been done related to this suggestion because the authors don’t know what specific phrases the reviewer was referring to:

  1. The first and second authors have star marks; however, Dr. Salinas, M. R is the only corresponding author, right?

The first two authors contribute equally to the present work and Salinas M.R. is the corresponding author. The mistake related to the marks has been corrected for a better understanding.

  1. The last author, Llorens and S, is wired, please correct it.

The name of the las author (Llorens, Silvia) has been corrected.

  1. In the abstract, try to use full name instead of “MTT”.

The full name related to the MTT has been included as suggestion.

  1. Page 2, paragraph 4, Vibrio fischeri should be italic. Please check the whole manuscript to avoid this kind of mistake.

It has been changed and checked throughout the manuscript.

  1. Page 5, line 2, R2, 2 should be superscript.

It has been corrected.

  1. Page 5, 2.4.2 MTT assay. Paragraph 2, line 4, 2 x 103 cells/well, I think 3 should be superscript.

It has been corrected.

  1. Page 9, line 1, is it possible to the note to previous page, so that the table is only separated in two pages instead of three pages.

Table 1 has been slightly modified according to the reviewer's suggestion.

  1. Pages 13-14, please try to arrange Figure 3 in the same page.

It has been changed accordingly.

  1. Page 15, the final sentence of paragraph 2. Authors choose 3T3-L1 to do the toxicity assay, and why didn’t you use Caco-2 or Hep-G2 cell?

Reviewer’s suggestion is interesting, especially the Caco-2 cell line, which is used to determine bioaccessibility of compounds, however these are tumourogenic lines. Authors have preferred a nontumorigenic cell line for viability assays because does not display an altered cell death potential, avoiding any interference with the testing outcomes.

  1. Furthermore, do authors discover any anti-obesity effect on the vine-shoot extract treated 3T3-L1 cell?

It is a very interesting topic. However, the adipogenesis assays exceeded the main objective of the research, but it can be taken into account for future works.

  1. Among references 5-7, the first author maybe the same person, but the formats are different.

Reviewer is right but, in her first publication (Cebrián, C., Sánchez-Gómez, R., Salinas, M. R., Alonso, G. L., & Zalacain, A. (2017). Effect of post-pruning vine-shoots storage on the evolution of high-value compounds. Industrial Crops and Products, 109(September), 730–736), Cristina Cebrían-Tarancón author used only the first surname. For reference 7, this aspect has been corrected.

  1. To me, reference 7 is incomplete. It may be a thesis or published in a meeting. Authors should provide more details.

Reviewer is right and reference 7 is related to the Cebrían-Tarancón’s Thesis. These details have been included.

Reviewer 3 Report

This is a very interesting research topic. My comments are as follows.

  1. A minor suggestion: Perhaps authors can use a colon (:) to connect two sentences.
  2. The first and second authors have star marks; however, Dr. Salinas, M. R is the only corresponding author, right?
  3. The last author, Llorens and S, is wired, please correct it.
  4. In the abstract, try to use full name instead of “MTT”.
  5. Page 2, paragraph 4, Vibrio fischeri should be italic. Please check the whole manuscript to avoid this kind of mistake.
  6. Page 5, line 2, R2, 2 should be superscript.
  7. Page 5, 2.4.2 MTT assay. Paragraph 2, line 4, 2 x 103 cells/well, I think 3 should be superscript.
  8. Page 9, line 1, is it possible to the note to previous page, so that the table is only separated in two pages instead of three pages.
  9. Pages 13-14, please try to arrange Figure 3 in the same page.
  10. Page 15, the final sentence of paragraph 2. Authors choose 3T3-L1 to do the toxicity assay, and why didn’t you use Caco-2 or Hep-G2 cell?
  11. Furthermore, do authors discover any anti-obesity effect on the vine-shoot extract treated 3T3-L1 cell?
  12. Among references 5-7, the first author maybe the same person, but the formats are different.
  13. To me, reference 7 is incomplete. It may be a thesis or published in a meeting. Authors should provide more details.

Author Response

Manuscript ID: FOODS-1226045

RESPONSE TO REVIEWER 3 AND CHANGES WE HAVE MADE TO THE MANUSCRIPT

Dear Authors,

my congratulations on excellent work and presentations of your results. There is just one little thing:

Authors would like to thank for the comments you have made on our paper.

On the page 7 of 17 you state that all varetis have been cultivated in the same way and that the soil provide a different mineral profile. It would be useful information if you could provide (in Material and methods chapter) more information about the soils and growing conditions, as well as about the way of growing in briefly.

The growing conditions, equal for the three varieties, have been added according to the reviewer's suggestion (line 113-115, page 3). Since the last ones were the same, the differences observed in the mineral composition were attributed to the soil, but it was only suggested, since soil analysis were not carried out in the study.